

# sandbox – Creating and Analysing Synthetic Sediment Sections with R

Michael Dietze[1], Sebastian Kreutzer[2,3], Margret C. Fuchs[4], and Sascha Meszner[5]

[1]GFZ German Research Centre for Geosciences, Section 5.1 Geomorphology, Potsdam, Germany
[2]Geography & Earth Sciences, Aberystwyth University, Aberystwyth, Wales, United Kingdom
[3]IRAMAT-CRP2A, UMR 5060, CNRS-Université Bordeaux Montaigne, Pessac, France
[4]Helmholtz-Zentrum Dresden-Rossendorf, Helmholtz-Institute Freiberg for Resource Technology, Freiberg, Germany
[5] JENA-GEOS-Ingenieurbüro GmbH, Jena, Germany

**Correspondence:** Michael Dietze (mdietze@gfz-potsdam.de)

**Abstract.**

The majority of palaeoenvironmental information is inferred from proxy data contained in accretionary sediments, called geo-archives. The validity of proxy data and analysis workflows are usually assumed implicitly, with systematic tests and uncertainty estimates restricted to modern analogue studies or reduced-complexity case studies. However, a more generic and

consistent approach to exploring the validity and variability of proxy functions would be to translate a given geo-archive into a model scenario: a "virtual twin". Here, we introduce a conceptual framework and numerical toolset that allows the definition and analysis of synthetic sediment sections. The R package `sandbox` describes arbitrary stratigraphically consistent deposits by depth-dependent rules and grain-specific parameters, allowing full scalability and flexibility. Virtual samples can be taken, resulting in discrete grain-mixtures with well-defined parameters. These samples can then be virtually prepared and analysed,

for example to test hypotheses. We illustrate the concept of `sandbox`, explain how a sediment section can be mapped into the model and, by focusing on an exemplary field of application, we explore universal geochronological research questions related to the effects of sample geometry and grain-size specific age inheritance. We summarise further application scenarios of the model framework, relevant for but not restricted to the broader geochronological community.

## 1 Introduction

Information about the evolution of earth-surface dynamics beyond the timespan of instrumental records is predominantly gathered from sedimentological deposits, serving as hosts of proxy data. Proxies are based on the presupposition that a specific sediment property is representative of an unknown environmental variable or can be unequivocally converted into such. For instance, the grain-size distribution of a sample is supposed to reflect the sediment transport processes, the isotopic composition of fossils represents precipitation or temperature, charcoal occurrence indicates human activity within a landscape, or

trapped charges in minerals denote a depositional age. The validity of proxies is usually an assumption based on conceptual relationships, modern analogue data, or physical principles. Further implicit assumptions arise from practical and methodological constraints, such as minimal post-depositional alteration, representative sampling, appropriate sample preparation and





measurement, and robust estimation of uncertainty ranges. All these preconditions are typically assumed or at least considered to be of generic validity. Developing tests of this assumed validity in the concrete context of a sedimentary deposit would allow

quantifying the level of confidence one can put into the proxy function utilised in a study.

Numerical modelling of the earth surface processes that contribute to terrestrial sediment cascades has reached an advanced level throughout the last few decades (Willgoose et al., 1991; Schoorl et al., 2000; Tucker et al., 2001; Lowry et al., 2013; Hobley et al., 2017). Yet, the commonly utilised landscape evolution models almost exclusively focus on specific parts of that sediment cascade, such as weathering, erosion, and material transport processes, or at least have model-specific strengths and

weaknesses in representing elements of this process phalanx. By contrast, the formation of sediment deposits, hence the actual carriers of the proxy information generated by the succession of processes, is rarely touched. Most often, sediment is simply flushed out of the terminal node or pixel of the modelled area, or deposition is reduced to the pure formation of geometric bodies (e.g., Lowry et al., 2013). In light of the importance of sedimentary deposits for palaeo-environmental research and predictive models of future dynamics that require long and robust information on past environmental conditions, it appears that

the host material angle of our palaeo-environmental picture is significantly understudied from a numerical perspective.

Describing the entity of a sedimentary deposit by a model would include a geometric description of the entire body (width, length, depth) as well as a thematic description of its constituents (e.g., voids, grains and their geometrical, mineralogical, or chemical composition, bulk water content or organic material concentration) by using a vast number of parameters. This is a challenge not easily tackled by any model. As an example, describing a ten metre tall and one by a one metre wide

column of loess would require describing as many as $10^{14}$ single grains, and each by a series of parameters. Depending on the research question, one might reduce the geometric dimension of the deposit and thus the number of individual constituents to describe. Likewise, it is possible to limit the number of parameters used to define each constituent. However, the general challenge remains. An alternative to this geometric and parametric reductionist approach is a model not at the scale of its discrete constituents but one with model-wide (general) rules that describe the properties of potential constituents at any given

location within the sedimentary deposit.

Ideally, such a model is transparent, open and flexible throughout. Transparency means that a user should have access to and gain insight into the model's structure and algorithm definition. Open means that the code is modifiable, and its functionality can be extended. Flexible means that the model can be used for many different purposes, covering a range of spatial and temporal scales, working with fewer and/or other than the default parameters, and addressing a series of research questions.

Here, we introduce the R package `sandbox`, a novel framework to create virtual, rule-based sediment deposits. We explain the concept and structure of `sandbox` along with a step-by-step description of how to map out a "real world" loess section in a model. We illustrate different fields of application based the created example data set. While we focus on geochronometric data for palaeo-research, various other application scenarios are imaginable, and hence, a suitability discussion of `sandbox` for other research questions beyond our modelling study will close our contribution. The SI contains an extensive tutorial to

the package along with all code used to create the figures of this article.





## 2 Philosophy and structure of sandbox

`sandbox` is a free and open framework to build and analyse virtual sediment sections in R (R Development Core Team, 2021). Upon the publication of the manuscript, the package will be made available at the Comprehensive R Archive Network (CRAN), ready for seamless installation across different computer platforms. Currently, the developer version is available on

GitHub (https://github.com/coffeemuggler/sandbox/). The term framework implies that `sandbox` is not tailored to a specific task but instead provides methods (R functions in our case) to use the tool in different scenarios. Users can reduce or expand the default range of parameters applied to describe the constituents of a sediment section. Due to the design of the R language, new functions can be added with a minimum of formal constraints to easily to exploit the virtual sediment section for user-specific purpose.

To make no mistake, `sandbox` is essentially a one-dimensional model. It describes the geometry of a sedimentary deposit only by its depth while assuming infinite width and length. Boundary conditions are treated irrelevant apart from the distance to surface. This reduction to one dimension restricts the applicability to problems that do not require a topological or geometric parameterisation of the modelled constituents (i.e., grain-to-grain relationships). Although it would be possible to extend parameterisation in lateral directions (2D or 3D), the additional effort to define parameters in two more spatial directions would

be considerable and is beyond the scope of this distribution. Furthermore, it is generally assumed that concordant accretionary sedimentary deposits vary predominantly with depth, the direction in which the material accretes with time, and should have similar properties in lateral directions – an assumption that forms the foundation of the stratigraphic principle and the correlation of sediment sections. This simplifies real-world conditions where deposits can vary in all directions due to the spatially and temporally non-uniform action of geomorphic processes. However, such occasions could still be handled with `sandbox`

if a user explicitly defines two distinct sediment sections that will be interpreted as if they would be spatially separated from each other.

`sandbox` has a parametric (termed rule-based from here on) and probabilistic design. Changes in sediment properties with depth are expressed as exact depth-dependent rules that define how bulk material parameters behave at any given section depth. In contrast, the actual realisation of a parameter value is set in a probabilistic way, i.e., a parameter is expressed by

a probability density function, whose parametric description changes with depth as defined by the rules. Thus, `sandbox` accounts for both uncertainty (reflecting the inability to determine, for example, the exact chemical composition of a sample) and strict user-imposed constraints on the model behaviour (reflecting the need to have the model behaving exactly as defined by the hypothesis).

`sandbox` allows not only to build synthetic sediment sections but also to (virtually) sample them, prepare the samples, mea-

sure them and work with the synthetic results as with real-world measurement data. Hence, it allows creating a "virtual twin" of a real geo-archive. The package contains several functions for particular tasks. For example, the function `make_Sample()` generates a finite number of sediment particles based on the rule-controlled parameters together with information on the sampling depth and sample container geometry. Thus, the `make_Sample()` function marks the transition from the probabilistic





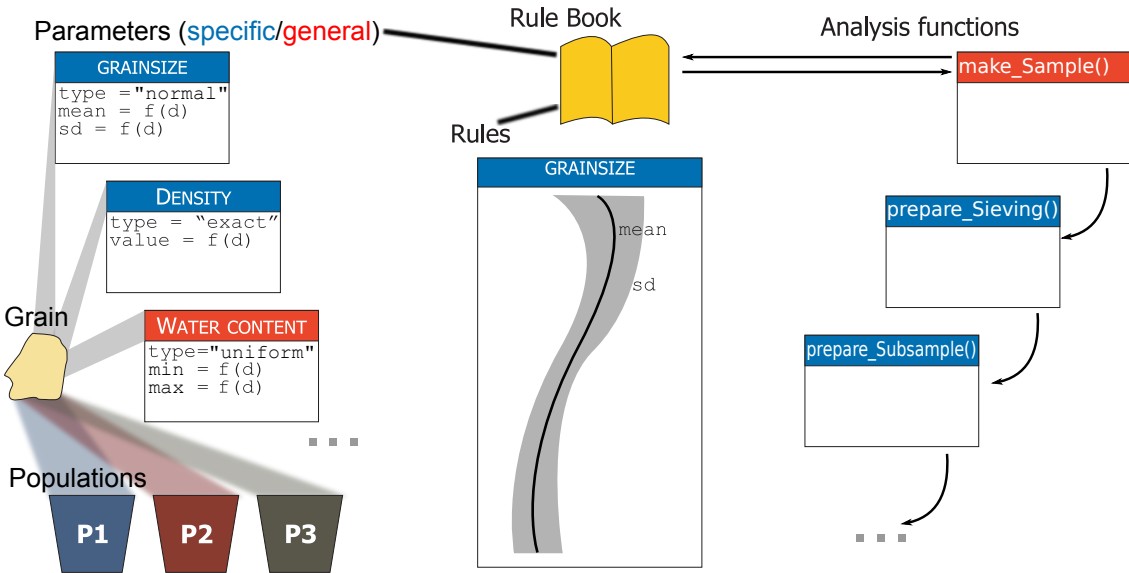

**Figure 1.** Concept of the model sandbox. Grains are the atomic elements of the model. They are drawn from populations and assigned parameters, which in turn are controlled by rules. Both rules and parameters are stored in rule books that act as coherent reference objects. Preparation functions are used to virtually process samples that are generated based on rule books.

and rule-based realm of `sandbox` to the discrete data realm. Such a set of sediment particles can then be passed to further

preparation functions (e.g., `prepare_Sieving()`) to simulate entire analytic processing chains.

Currently, `sandbox` focusses on the broader context of luminescence dating (e.g., Aitken, 1998), but it can be extended easily to other applications such as geochemical or mineralogical fingerprinting. Hence, if there are no pre-built functions available, it becomes necessary to define new preparation functions, a task that is achievable with deliberately low effort and programming experience in a language like R (R Development Core Team, 2021). Understanding `sandbox` (Fig. 1) requires

some key terms used in the modelling environment:

**Population**. A population is the most basic, coherent element of the entire model. A population is a set of sediment grains with common characteristics. All grains from one population share the same (range of) properties of certain parameters, such as grain size, depositional age or mineralogic composition.

**Grain**. Grains are the atomic elements of the model. They are always sampled from populations and described by a set of

parameters. Each population has a defined probability of occurrence, which is defined as a parameter.

**Parameter**. Parameters are used to describe populations and, hence, sediment grains drawn from these populations. Parameters can be seen as the "thematic" definition of a virtual sediment deposit. There are two major groups of parameters: general and specific. General parameters are depth-dependent sediment descriptions regardless of the population the grains are sampled from. Examples of general parameters are water content and external dose rate (ionising radiation per time unit). Specific





parameters describe sediment grains with respect to the population to which a grain belongs. Hence, for each population, there is another parameter definition. Examples are grain size, element or mineral constituents and specific density.

**Rule**. Rules describe how parameters change with depth. Rules can be regarded as the "spatial" definition of a sediment deposit. Rules are defined as interpolation functions based on a discrete number of parameter-depth relationships. The default interpolation function is a spline. This interpolator can easily describe a constant behaviour, a linear trend and further, local

trends in parameters. Splines are also meaningful smoothening interpolators between parameter-depth value pairs, a characteristic that can be seen as a pro or a con. Currently, stratigraphic discontinuities are not explicitly accounted for.

**Rule book**. A rule book is the combination of parameters ("thematic" definition) and rules ("spatial" definition) to one coherent reference book. A rule book ultimately comprises the definition of the entire virtual sediment section and generates individual samples. There is an empty rule book available by default. A user can modify a rule book's content at any time.

**Analysis function**. Once a rule book defines a virtual sediment deposit, it can be "exploited" using the pre-selection of available functions, for example, by generating sets of samples with `make_Sample()`. These samples can then be subject to additional analysis functions of the package, such as `prepare_Sieving()` and `prepare_Subsample()`. All these `prepare`-functions use information stored in each grain.

## 3 Materials and methods

### 3.1 Available functions

After installing the package (`install.packages("sandbox")`) and loading it (`library("sandbox")`), the package functions are available to the user. To start from scratch, it is necessary to create a new empty rule book, which can then be expanded by adding rules and parameters. A new rule book can be created by the function `get_RuleBook()`, using the default keyword `book = "empty"`.

```
book <- get_RuleBook(book = "empty")
```

This will generate a list object with all principal elements required to define a virtual sediment section: a book name (`$book`), a true age definition (`$age`), and the definition of population likelihoods (`$population`), grain-sizes (`$grainsize`), packing density (`$packing`) and specific grain density (`$density`). True age means that underlying all concepts of `sandbox` there is the assumption of an actual initial age for each grain depending on its depth. Grain-sizes are defined on a $\phi$-scale

throughout ($\phi = -log_2(\frac{D}{D_0})$, with $D$ the diameter in $\mu$m and $D_0$ the reference diameter1000 $\mu$m) to account for the nonnormal distribution of these data (Krumbein, 1937). Packing density describes the ratio between compound sample volume and the volume of solid particles in that sample. For regular spheres in a 3D space, the close-packing density cannot exceed 0.74 (Hales, 1992). For natural soil material, the packing density is usually around 0.3 to 0.6 (Blume et al., 2010). The packing density becomes relevant for `sandbox` when taking virtual samples by volume or further volume-based processing steps. The

specific grain density (2.65 g cm$^{-3}$ for quartz) is needed to define grain masses.



The function `add_Population()` allows adding other grain populations to a rule book, which by default only has one. Populations can be added at any time, and all specific rules of the rule book will be updated for the respective number of additional populations. The function requires specifying the rule book to be updated and providing the number of populations to add. The code below adds one population and stores the changes as a new rule book.

```
book2 <- add_Population(
      book = book,
      populations = 1)
```

Using `add_Rule()` allows to expand the range of application and add, for example, information about the chemical or mineralogical composition of a sediment section. The new rule will automatically add the corresponding new parameter. The
function requires specifying the rule book to be changed (`book`), a name (`name`) for the new rule and corresponding parameter, whether it is a specific or general rule (`group`), and how the resulting parameter is allowed to vary (`type`). Possible variation types are `type = "exact"` (no variation), `type = "normal"` (variability according to a normal distribution defined by additional rules of mean and standard deviation), `type = "uniform"` (variability according to a uniform distribution defined by minimum and maximum values), and `type = "gamma"` (variability following a Gamma distribution defined by
its shape and scale parameters as well as an offset constant). Depending on the type of variability, the function will add the required parameters (value, mean, sd, min, max, shape, scale, offset) to the rule book. To add, for example, a rule that defines a uniformly varying pH value for all populations, the following code is needed:

```
book3 <- add_Rule(book = book,
                  name = "pH",
155               group = "general",
                  type = "uniform")
```

The function `set_Rule()` allows defining the actual rules of the rule book. An empty rule book just contains the templates of required rules (five in total). These templates need to be filled with proper definitions. This is the main purpose of `set_Rule()`. Depending on how a rule defines the parameter variability, one needs to provide different information along
with their respective depth intervals to establish the right interpolation function. To define the rule for grain depositional ages, one needs to define a list that contains the depth intervals for the corresponding true age information and assign this to the rule book. To assign grain density rules, allowing for variability around a mean with a given standard deviation, one needs to create a nested list, one for each population, containing the means and standard deviations at the corresponding depth intervals. The below example will first define the rules as a one-metre depth interval, with a linear age increase of 1 ka per metre. Then, the
density for the population (`P1`) is defined as 2.5 g cm$^{-3}$ on average, but with a depth-dependent standard deviation. Finally, the grain packing density is set to 0.5 without scatter throughout the sediment section.

```
## describe rule definitions
depth <- list(c(0, 1, 2, 3))
```




```
     age <- list(c(0, 1000, 2000, 3000))
density <- list(P1 = list(mean = c(2.5, 2.5, 2.5, 2.5),
                                sd = c(0.0, 0.1, 0.2, 0.0))))
     packing <- list(P1 = list(mean = rep(0.5, 4),
                                sd = rep(0, 4)))

## assign age rule
     book <- set_Rule(book = book,
                      parameter = "age",
                      value = age,
                      depth = depth)
     ## assign density rule
     book <- set_Rule(book = book,
                      parameter = "density",
                      value = density,
185                  depth = depth)

     ## assign packing rule
     book <- set_Rule(book = book,
                      parameter = "packing",
190                  value = packing,
                      depth = depth)
```

    `make_Sample()` is a special function used to turn the essential information of a rule book into a discrete set of $n$ grains. For this, the function requires the arguments `book` (the rule book used to define the sediment section), `depth` (defining the centroid depth where the sample is created), and `geometry` (defining the geometrical shape of the sample container).

Currently, two types of sample containers are implemented: `"cuboid"` and `"cylinder"`. Depending on which of the two container shapes is used, further input is needed regarding `height`, `width`, `length` and `radius`. The function first estimates the number of grains needed to fill the volume of the sample container. To do so, a test set of 1000 grains is sampled using uniformly distributed random depth values inside the depth interval defined by the container dimension. With the discrete grain depths at hand, the rule book is queried to get information about the likelihood with which grain may come from one of

the defined populations. The population value is then used to generate grain-specific information about the depth-dependent mean and standard deviation of the grain diameter and packing density to calculate the volume filled by this subset of 1000 grains. The percentage of the subset volume gives an estimate of the number of grains required to fill the entire sample volume. The function will add 10 % to this number to account for inaccuracies due to the small test sample size and then generates a





full set of grains to fill the sampling container following the above-described workflow and ultimately removes the about 10 %
superfluous grains. The function output is a `data.frame` object with all grains, each described by the parameters contained
in the rule book. The following code snippet shows an example (output slightly modified for illustrative purpose) of a 1 cm³
large cubic sample created by `make_Sample()`, containing 491 grains each described by eight parameters:

```
sample_1 <- make_Sample(book = book,
depth = 1,
geometry = "cuboid",
height = 0.01,
width = 0.01,
length = 0.01)

str(sample_1)

'data.frame': 491 obs. of  8 variables:
$ grains    : num  1 2 3 4 5 6 7 8 9 10 ...
$ d_sample  : num  0.997 0.999 0.999 1.002 1 ...
$ population: num  1 1 1 1 1 1 1 1 1 1 ...
$ age       : num  997 999 999 1002 1000 ...
$ population: num  0.997 0.999 0.999 1.002 1 ...
$ grainsize : num  2.4 2.12 1.27 1.57 1.58 ...
$ packing   : num  0.5 0.5 0.5 0.5 0.5 0.5 0.5 0.5 0.5 0.5 ...
$ density   : num  2.56 2.45 2.5 2.62 2.53 ...
- attr(*, "package")= chr "sandbox"
```

The element `grains` contains the IDs (`grains`) and (`d_sample`) the depth of each grain. The element `population`
specifies to which population a grain belongs (all 1 since we only used one population). The second (`population`) entry
is just a necessary entry due to the sampling process and can be ignored. The remaining four elements (`age`, `grainsize`,
`packing`, `density`) correspond to what has been defined in the rule book.

With `prepare_Sieving()`, one can simulate the physical sieving of a sample. The function requires the arguments
`sample` (the sample object to be processed) and `interval` (sieve interval in $\phi$ units). Based on this information, it will
remove all grains from the sample object that do not fall into the sieve intervals and return the updated data set.

Splitting a bulk sample into a set of subsamples is performed by the function `prepare_Subsample()`. This can be done
by splitting a sample into a defined number of equally large subsamples (specified by the argument `number`), by creating
subsets of a defined volume (specified by the argument `volume`), or by creating subsamples defined by sample weight (speci-
fied using the argument `weight`). In the latter two cases, the remainder of the bulk sample that does not allow filling the last



subsample will be rejected. The volume option accounts for the packing density, and the weight option accounts for the specific density of the sample grains.

A special kind of subsampling is performed by the function `prepare_Aliquot()`. Aliquots are defined in luminescence analysis as sample subsets that compose a monolayer of sediment, fixed onto small metal discs, supplied to the measurement device. The function mimics this typical workflow step and requires the specification of the aliquot disc size containing the grain monolayer and a packing density of the grains on that disc, usually 0.65. Note that this value differs from the original packing density value used to define the rule book.

Finally, the package contains a convenience function `convert_units()`. This function can be used to convert grain-size units between the metric and the $\phi$ scale. Note that the package also contains further function `add_Parameter()`, which is a helper function used by `set_Rule()`, and not for direct usage.

### 3.2 The loess deposit Gleina

We have defined a virtual sediment section using measured data from a real-world loess deposit for the example cases discussed
here. The section is in a former brickyard in the Saxonian Loess Region, eastern Germany (Meszner et al., 2013). The entire sequence has a thickness of about 17 m spanning over 26 distinguishable stratigraphic layers with a Holocene Luvisol at the top and an ochre sand base (Meszner et al., 2011). The loess record gained regional popularity for its prominent, intensely coloured interstadial soil complex called "Gleina soil complex". While the site is an important type locality of superregional relevance, parts of its interpretation are still an open debate. Most importantly, Meszner (2015) called for caution regarding
the interpretation of the "Gleina soil complex" as a compound feature (ca. 40–80 ka, Zech et al. 2017). Rather, it should be considered as two separate layers with a hiatus of some 30 ka. For this site, we can access a detailed granulometric dataset (Meszner et al., 2021) along with a geochronological framework (Zech et al., 2017), both suitable in the context of this study.

The grain-size distribution of all 42 samples of the loess section were measured with a Horiba LA950 laser particle sizer, providing 98 grain-size classes. About 0.5 to 1.5 mg air-dried and homogenised material have been treated with 10 % HCl
for 24 hours and subsequently treated with 40 % $H_2O_2$ for 72 hours. Each sample has been measured for 5 s with ultrasonic excitation for 10 s in the device to disaggregate particles mechanically. We used the Mie scattering theory with a refraction index of 1.55 and an absorption index of 1.33. The median distribution of 10 consecutive measurements per sample has been exported for further analyses.

### 3.3 EMMAgeo as auxiliary tool

To convert the quasi-continuous grain-size distributions into discrete populations, i.e. parametric descriptions (mean and standard deviation) of grain-size rules for `sandbox`, we unmixed the dataset using the R package `EMMAgeo` v0.9.6 (Dietze and Dietze, 2016, 2019). This package allows end-member modelling analysis (EMMA) of grain-size data sets and uses principles of eigenspace analysis to describe grain-size distributions as a linear combination of end-member loadings and scores. Loadings are the fundamental, genetically interpretable grain-size distributions inherent to all grains. They can be interpreted
in terms of discrete sediment sources, transport pathways and/or transport processes. Scores depict the contribution of each





loading to each sample. Thus, scores can be considered as a description of the relevance of a transport process for a given sample. In the context of this study, loadings refer to the grain-size distribution of particular populations (parameter definition), and scores refer to the depth-dependent likelihood of a population to be sampled (rule definition).

In the case of the Gleina loess section, EMMA has been used in the deterministic mode with a manually defined number of
three end-members ($q = 3$). The weight transformation limit has been set to zero ($l = 0$). Both parameters were not extensively optimised because the goal of this analysis step was not to achieve the best possible representation of the measured data set but to generate input data for `sandbox` in general agreement with a real data set. The resulting loadings have been approximated with log-normal distribution functions (i.e. normal distribution functions in the $\phi$ space) to get the best fit values of mean and standard deviation for each of the three populations. The scores have been appended to the existing parameter data set of the
Gleina loess section.

### 3.4   Mapping a deposit into sandbox

All analytical data of the Gleina loess section (Meszner et al., 2011), including the reanalysed grain-size data and end-member scores, have been linearly interpolated to equal intervals of 25 cm, starting at 0.5 m depth and ending at 10.75 m depth (see SI for details on the data set). We have used the fine-grain luminescence ages (Zech et al., 2017) to build an interpolated age-
depth relationship as true age-depth information, despite potential ambiguities, just for the sake of simplicity and to serve as an example.

We have created a new empty rule book (`gleina`) and added two more grain populations to the default one, in agreement with the results of the end-member modelling analysis. The relative contributions of the three end-members (i.e., their depth-dependent scores, `EM_scores`) have been used as population probabilities and added to the rule book. The parametric
approximations of the end-member grain-size distributions (`EM_gsd`) were added similarly. We set the population-specific packing densities to 0.7 for the coarse-grained end-member, 0.6 for the medium and 0.5 for the fine end-members, each with a standard deviation of 0.01. Grain specific densities were set to $2.65 \pm 0.01 \, \mathrm{g \, cm^{-3}}$ for all populations, imposing predominantly quartz minerals. The following code snippet is a one-to-one version of this descriptive text.

```
## load the measurement data
X <- read.table(file = "gleina_measurements.txt")

## convert cm to m, get number of records
X$depth_int <- X$depth_int / 100
n <- nrow(X)

## create empty rule book
gleina <- get_RuleBook(book = "empty")
```



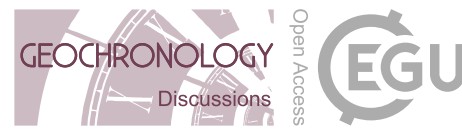

```
      ## add two further populations
gleina <- add_Population(book = gleina, populations = 2)

      ## assign rule definitions to lists
      depth <- list(X$depth_int)
      age <- list(X$age_int)
EM_scores <- list(list(X$EM_1),
                        list(X$EM_2),
                        list(X$EM_3))
      EM_gsd <- list(list(mean = rep(6.38, n), sd = rep(0.9, n)),
                     list(mean = rep(4.69, n), sd = rep(0.5, n)),
315                  list(mean = rep(4.29, n), sd = rep(0.5, n)))
      EM_packing <- list(list(mean = rep(0.7, n), sd = rep(0.01, n)),
                         list(mean = rep(0.6, n), sd = rep(0.01, n)),
                         list(mean = rep(0.5, n), sd = rep(0.01, n)))
      EM_density <- list(list(mean = rep(2.65, n), sd = rep(0.01, n)),
320                      list(mean = rep(2.65, n), sd = rep(0.01, n)),
                         list(mean = rep(2.65, n), sd = rep(0.01, n)))

      ## add rule definitions
      gleina <- set_Rule(book = gleina, parameter = "age",
325                    value = age, depth = depth)
      gleina <- set_Rule(book = gleina, parameter = "population",
                         value = EM_scores, depth = depth)
      gleina <- set_Rule(book = gleina, parameter = "grainsize",
                         value = EM_gsd, depth = depth)
gleina <- set_Rule(book = gleina, parameter = "packing",
                         value = EM_packing, depth = depth)
      gleina <- set_Rule(book = gleina, parameter = "density",
                         value = EM_density, depth = depth)
```

## 3.5 Application examples and parameterisation

To illustrate the basic functionality and potential applicability of sandbox, we investigate a set of research questions. We
provide the R code that has been used to implement each of these tests in the SI. The questions are as follows:



1) **How does the sample container geometry impact the age scatter?** Container geometry means that we have inspected the differences between cylindric and cuboid sample containers, all with the same volume. We simulated cylinders with 10 mm diameter, cubes of 10 mm width and height and 0.8 mm length, and cuboids of 20 mm, 40 mm, and 80 mm width and 5 mm, 2.5 mm, and 1.25 mm height, respectively. While the real-world applicability of such container geometries is limited, the test stresses the influence of sampling depth intervals, including minimal values. It can also be interpreted as mimicking manually extracting a thin layer of sediment without using a sample container. Cuboid container lengths were set to 0.8 mm throughout. Note that we can work directly with the sampled material without any further preparation steps. Thus, small sample volumes are sufficient. The virtual sampling depth for the test was set to 5 m, using the depth-interpolated true ages of the sampled grains as a direct proxy for age scatter.

2) **What is the effect of sample container size on age uncertainty?** Here we test different cylinder diameters for different profile depths of the Gleina loess section. We have sampled the virtual Gleina section at 1 m intervals, using containers with diameters ranging from 0.5 cm to 50 cm, keeping the volume constant at 0.5 cm$^3$. Again, 0.5 cm and 50 cm wide containers are far from reality. However, they define a safe lower and upper limit of possible cases and manually collected samples.

3) **What kind of standard luminescence age bias is expected due to grain-size interval processing if the three components building the loess section have different bleaching probabilities?** For this test, we have added a new specific rule (`inherited`), which defines the inherited age in years for each population. For the coarse-grained population (end-member 3), we assumed a poor bleaching likelihood and thus a uniformly distributed random age inheritance within the arbitrarily chosen range of 0 years to 5000 years. For the two other populations, we imposed uniform random inheritance ages between 0 years and 200 years. We collected samples every 0.5 m using a 5 cm cylinder, sieved the sampled material for the typical coarse grain (90–200 $\mu$m) and fine grain (4–11 $\mu$m) fraction (e.g., Kreutzer et al., 2012a) and calculated the mean age composed of the true deposition age and the inheritance from each grain. In addition, we have prepared three additional data sets, this time adjusting the limits of the virtual sieve to isolate each of the three end-members as good as possible (see Fig. 2 for intervals), in order to inspect the age differences inherent to the three different components that constitute the sediment section.

# 4 Results

## 4.1 End-member modelling analysis

Deterministic EMMA with three end-members ($q = 3$) and no weight limit transformation ($l = 0$) resulted in an overall R$^2$ of 0.74 (sample-wise R$^2 = 0.93$, class-wise R$^2 = 0.56$). The class-wise R$^2$ increased to 0.75 when omitting the low values in classes where the end-member loadings are close to zero due to only minor vol.-% values in the input data set, see Fig. 2. The three end-members were unimodal with $\phi$ modes at 6.38, 4.68 and 4.29 (12 $\mu$m, 39 $\mu$m and 51 $\mu$m). Secondary, artificial, peaks occurred below the main modes of the other end-members, as commonly encountered in EMMA (Dietze and Dietze, 2019). Hence, normal functions were only fitted to the primary modes. The best fits were reached for $\phi$ 6.38±0.9, 4.68±0.5 and 4.29±0.5 These values were used to feed the rule book with information about the grain-size distribution of the three





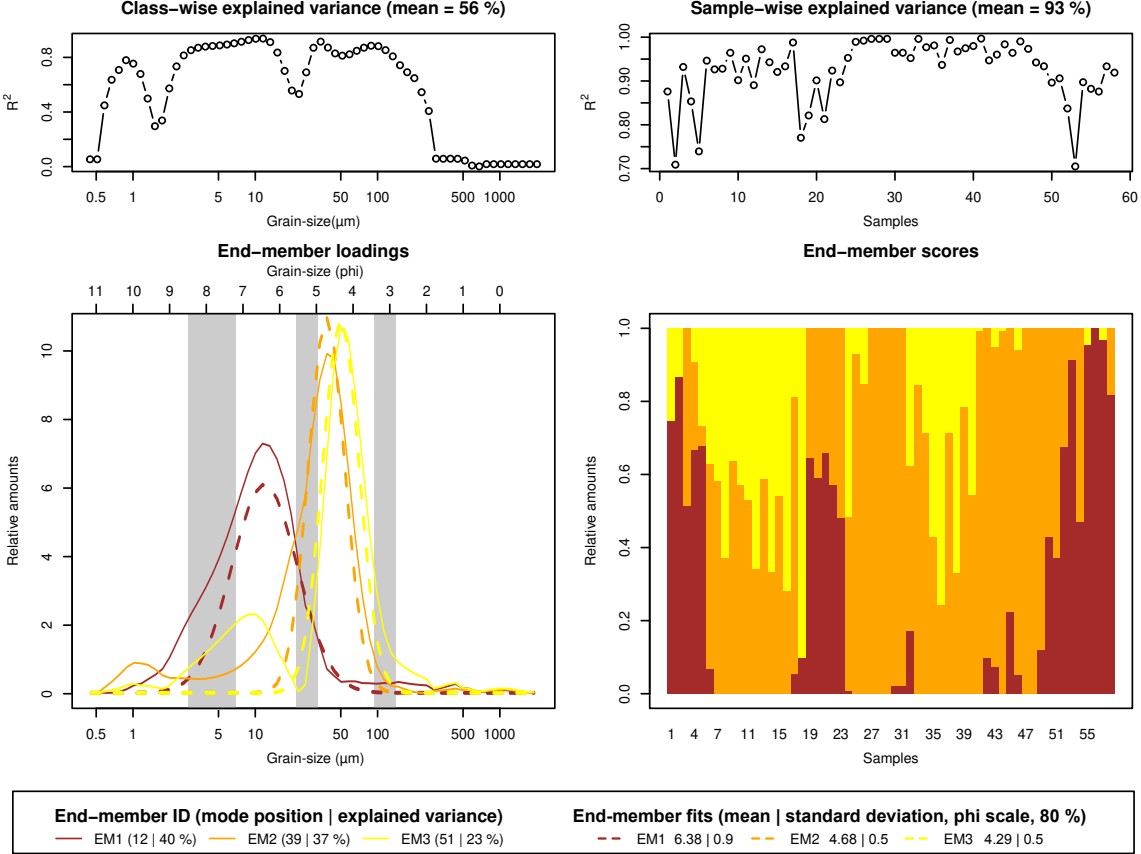

**Figure 2.** End-member modelling results of the measured grain-size data of the Gleina section. Dashed lines in the loadings plot show fitted log-normal distribution curves according to the parameters in the bottom legend panel. Grey shaded areas in the loadings panel depict grain-size intervals used for sieving to enrich individual end-members. Sample IDs in the scores plot denote samples from top to bottom.

populations (mean and standard deviation were set as constant, i.e., not varying with depth). The probability of grains to be

drawn from one of the three populations was defined by the respective end-member scores.

### 4.2 Effect of sample container geometry and size on age scatter

The sample container shape directly reflects the single-grain age distribution of the sampled material (Fig. 3). Cylinders produce a sinusoidal distribution shape of ages with a standard deviation of 3.7 years, while cubic containers produce a flat distribution with a standard deviation of 4.2 years. Note that the absolute scatter is an arbitrary number with no real-world equivalent. It

merely depends on the deposition rate and container size (see below). Along that line, as the cuboids become more elongated in the horizontal direction by factors 2, 4 and 8, the standard deviations of the flat age distributions decrease to 2.1 years, 1.1





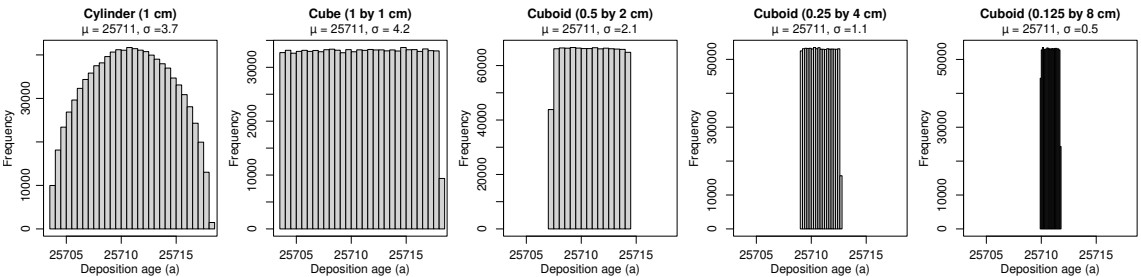

**Figure 3.** Effect of sample container geometry on scatter of sampled grain ages. X-axes all scaled to the same range.

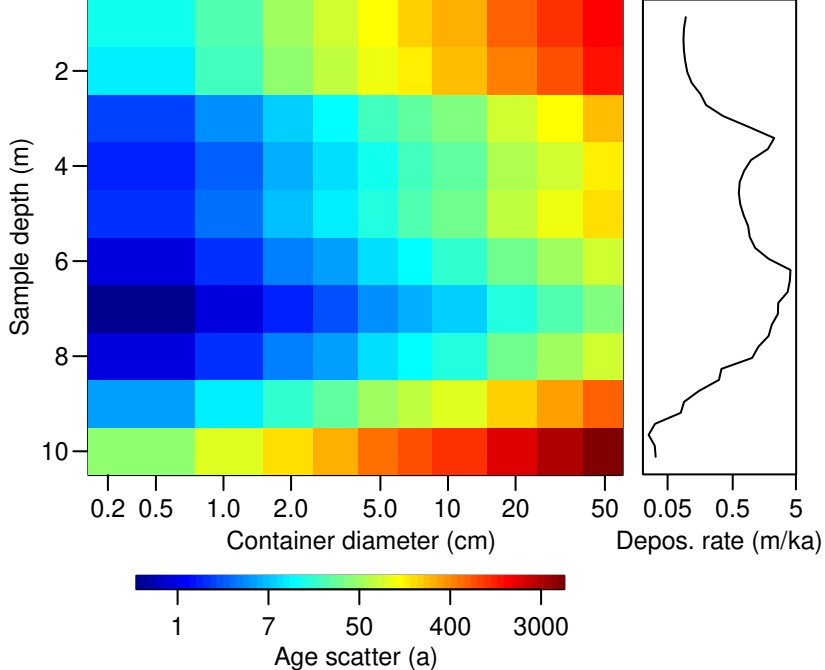

**Figure 4.** Effect of cylindric sample container size. Warmer colours indicate a higher age scatter.

years and 0.5 years, respectively. This purely geometric effect is also witnessed by the constant average age for all types of sampling containers.

Sample container size has a variable effect on the age scatter (Fig. 4), also depending on the deposition rate of the investigated
sediment section. In general, larger sample container sizes systematically increase the age scatter inherent to the sampled grains, following a linear relationship. However, the sampling depth modulates the overall scatter, which determines the sediment deposition rate here. For the deposition ranges of the virtual Gleina section, we found age scatter as high as 4624 years (using a 50 cm wide sampling depth interval) in the basal section with a deposition rate of 0.033 m ka$^{-1}$. More realistically, 5 cm wide





sampling containers still yielded an age scatter of 481 years. In the central parts of the section with deposition rates as high as
3.9 m per ka, that error reduced to 4 years and 43 years for container diameters of 5 and 50 cm, respectively.

### 4.3 OSL age bias due to preparation size ranges

The impact of age inheritance can range from marginal to significant, depending on the analysed grain-size fraction. Note that here we can ignore scatter in age and thus error bars in Fig. 5 because we implicitly know the true ages of the sampled grains and hence can focus completely on the systematic effects. Analysing the typically utilised coarse grain fraction (90–200 $\mu$m,
Fig. 5 a) can introduce a systematic mean age offset (difference between apparent and true age) of up to 2500 years (up to 10 %). Thereby, the offset is controlled by the relative contribution of the coarse-grained end-member to a sample. The result is a stratigraphically inconsistent age-depth relationship with four age inversions. When using the typically encountered size interval of the fine-grain fraction (4–11 $\mu$m, Fig. 5 b) to estimate average grain ages, the age offset is minimal, about 118 years on average. There are no age inversions visible in this size fraction. However, the age offset still correlates with the contribution
of end-member 1, from which a few grains still leak into the sieving interval.

When targeting grain-size intervals that specifically aim to isolate the three end-members inherent to the grain-size distribution of all samples, the coarse-grain end-member (Fig. 5 c) mimics the offset and stratigraphic inversion patterns of the coarse grain samples. The intermediate end-member (Fig. 5 d) shows similar trends to the coarse one but with less severe effects (800 years maximum offset). Finally, the fine-grain end-member (Fig. 5 e) shows an average age offset of 100 years (corresponding
to the imposed range of 0–200 years) without any relationship to the contribution of the coarse-grained end-member.

## 5 Discussion

### 5.1 Structure and implementation of sandbox

The proposed structure of `sandbox`, consisting of grains, populations, parameters, rules, and functions, allows to consistently define synthetic sections (virtual twins of sediment deposits) that can be used to pursue a series of research questions, for
instance the study of the effects of sampling container size and grain size related age inheritance on the expected age scatter. The latter one would be usually recognised as overdispersion, i.e., unexpected variance in the age distribution regarding a pre-defined central age.

Further extension of any given rule book is possible regarding additional populations, grain parameters and rule definitions. The available distribution functions to describe parameters and rules cover a significant range of use cases. Adding other
functions would require updating the R code of the package, either by a new package release or by editing the functions manually. Both is possible due to the R language constraints and because the package only uses standard R packages by default.

The fundamental assumption inherent to `sandbox` is a valid, true age-depth relationship. At face value, this assumption implies that a sediment section fulfils the stratigraphic principle. However, this enforced principle is implemented through a



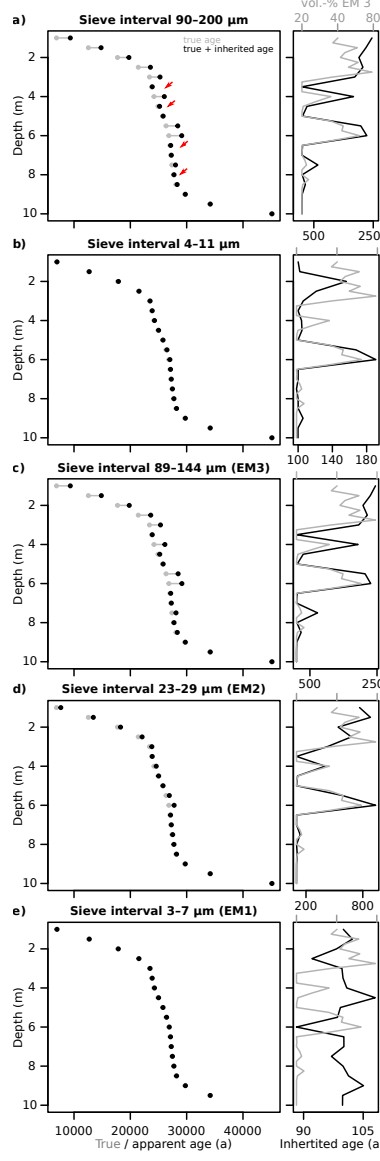

**Figure 5.** Age inheritance effects due to different sieve intervals of the modelled Gleina section. a) Typical coarse grain sieve intervals. b) Typical fine-grain sieve intervals. c–e) Optimised sieve intervals to isolate the three inherent grain-size end-members best possible. See Fig. 2 for interval definitions. The left plot panels show average true depositional ages (grey dots) and apparent measurement ages (black dots), composed of true and inherited ages per grain. Right plot panels show age inheritance (black lines) and contribution of end-member 1 (grey lines) as a function of depth. Red arrows mark age inversions.

spline function. Such an interpolator of discrete age-depth pairs works appropriately as long as there is steady accretion of





material with time. However, it reaches limits (and fails its purpose) when stratigraphic gaps need to be addressed. In such a case it is better to create two separate synthetic sediment sections, one above and one below the gap.

## 5.2 The virtual Gleina section

A cornerstone for subsequent analysis examples was translating the measurement-based description of the Gleina loess section
into parameters and rules for `sandbox`. End-member modelling analysis yielded meaningful results: the three end-members are predominantly unimodal (Fig 2). The secondary mode of EM 3 that emerges right below the main mode of EM 1 is a typical model artifact (Dietze and Dietze, 2019) and should thus not be interpreted as genetically meaningful. However, EM 2 has a suppressed secondary mode around 1 $\mu$m, which is statistically robust (high $R^2$ values) and does not interfere with a mode of any other end-member. Thus, this secondary mode may indeed represent a transport regime that contributed a primarily
bimodal grain-size distribution. It has been repeatedly reported that loess particles are transported not just as single grains of medium to coarse silt size but also as aggregates of smaller particles, either forming silt-sized agglomerates or adhering to such larger particles (Vandenberghe, 2013). In general, the three end-member loadings show the typical properties of Central European loess deposits (e.g., Bertran et al., 2016). The dominance of the coarse silt fraction – in the Gleina case, two distinct populations (EMs 2 and 3) that both contribute predominantly to the least pedogenetically altered depth intervals (Fig 2) –
and an additional contribution of fine silt, in the Gleina case, this is EM 1. Interestingly, none of the morphologically distinct palaeo-soil horizons (Meszner et al., 2011) exhibits remarkable and distinct contributions of clay-sized particles. Instead, it is EM 1 that dominates the distributions in the pedogenetically altered depth intervals.

## 5.3 Geometric sampling effects

The shapes of the employed sample containers have, in general, purely geometric effects on the age composition of each sample
(Fig 3): circular containers result in a sinusoidal age distribution and rectangular containers in a flat one. From a relative age scatter perspective, cylindric containers are preferential to cube-shaped ones of the same vertical extent because most grains are sampled from the desired target depth. Furthermore, flatter containers result in linearly decreasing relative age scatter. These findings may be rather obvious and serve as a simple example of how easily questions may be approached quantitatively yet systematically with `sandbox`.
Container size and shape become more relevant when the material deposition rate is considered. For high deposition rates, like those typical for loess environments (e.g., the central part of the Gleina section between 8 m and 3 m depth, Fig. 4), age differences among grains due to container size is small, a few years per cm container height. However, when section intervals with low loess deposition rates are sampled, such as the basal and top parts with more prominent pedogenic features, the age scatter can increase by several orders of magnitude simply because the grains in the container represent a larger range of true
ages. Hence, age scatter due to sampling is no artifact but actually represents the range of smapled grain ages. In the Gleina section that we use as an example, a standard luminescence sampling cylinder of 5 cm diameter can thus add an age scatter of several hundred years, regardless of the absolute depositional age.





While there are ways to minimise this sample container size effect, the documented practice in published articles seems to show that in most cases standard sampling containers are used. Reducing the age scatter may be accomplished by using flatter

and more elongated sampling containers or even extracting material from horizontally aligned slits carved into an outcrop when this is possible. This sampling procedure requires more manual adjustment and is thus prone to other shortcomings (e.g., sample contamination, light-shielding efforts). Nevertheless, we advocate that virtual sediment modelling is used in advance to estimate the expected age scatter effect for given sample container geometries if one has a prior order of magnitude estimate of the deposition rate, for example, based on stratigraphic relationships with other sections.

## 5.4    Sample population effects

In the age bias modelling exercise, we have explored how it is possible to simulate grain population-specific age inheritance effects and to which extent these can impact the resulting subsample ages when subsampling is achieved by sieving. Then, we have shown how one can attribute age inheritance phenomena to the underlying grain-size based populations identified by EMMA.

Real-world analogues of age inheritance could be poorly bleached grains, an effect that in many cases has been attributed to a transport process that exposes those grains to direct sunlight only randomly and for brief time intervals. Examples of such processes would be bedload or near bed transport in turbulent rivers or dislocation as rapid mass wasting processes or rapid transport by soil erosion (Fuchs and Owen, 2008; Fuchs et al., 2010). Typically, such specific transport processes tend to focus also the grain-size distribution of the material they carry (Weltje, 1997; Vandenberghe, 2013). Hence, this problem fulfils the

preconditions for end-member modelling analysis in a particular way. The technique allows isolating the transport processes due to their characteristic grain-size distributions. The resulting end-member loadings information can then be used to adjust the sieve interval limits for subsequent age determination analyses, to the extent that other analytical operations permit this.

We found that straightforward application of good practice, isolating the typically utilised coarse grain or fine grain fractions for luminescence dating, can lead to two very different age estimates between the fractions. Unfortunately, in principle, it is not

possible to tell which of the two is more correct than the other – apart from the fact that in our example, one of the age-depth relationships was stratigraphically consistent, whereas the other was not. This inability to identify the "correct" solution is due to the phenomenon of multiplicity: different mechanisms leading to the same, equivocal result. The two-grain size fractions are subject to different microdosimetric effects, and sample preparation work flows, both being potential causes of differences in the resulting depositional age estimates (e.g., Fuchs and Lomax, 2019). In addition, the two grain-size fractions are also subject

to different transport processes and depositional circumstances. These two classes of effects, methodological and transport dynamics, can affect the resulting age estimates in a cumulative and counteracting way. Hence, to at least account for the transport class of effects, we recommend applying EMMA before deciding on the grain-size fraction to use for subsequent age determination workflows. This approach does not only quantify the number and grain-size characteristics of the populations inherent to a set of samples but it also allows adjusting the grain-size fractions for age determination to ideally avoid overlapping

of end-members. In other words, the sample preparation process isolates only grains that belong to the same transport regime.





## 5.5 Potential further applications

This article's primary purpose is to introduce the synthetic sediment section modelling framework sandbox, particularly with emphasis on luminescence-based age determination. We have demonstrated how the framework can be modified in general. Thus, it is possible and encouraged to apply the package beyond such simple or rather specific examples. sandbox can also be used to pursue questions inherent to other radiometric age determination techniques, such as electron spin resonance, radiocarbon dating, detrital zircon dating, or even varved lake sediments or dendrochronology age models. When parameters are assigned for grains' mineralogical or chemical composition, further scientific questions can be approached. For example, from disciplines like provenance analysis (based on detrital zircon age distributions, mineralogical composition, or rare earth element concentrations). In the supplementary material of this article we provide an example about how sandbox can be linked with other R packages, such as RLumModel (Friedrich et al., 2016) and Luminescence (Kreutzer et al., 2012b).

Inverse problems (Zeeden et al., 2018) are another potential cross-topic field of application of sandbox. In many cases, there are no analytical solutions to link multi-parameter workflows to given sets of outcomes. Hence, one can only run large scenarios with different parameter combinations to identify the parameter space that can deliver plausible solutions. The sandbox framework provides the flexibility and efficiency needed to run many such scenarios for different questions.

A further independent field of application regards the definition of reference data sets, for example, to test age model approaches (e.g., Galbraith et al., 2005) or to explore the potentials and limitations of mixed-age distributions (Arnold and Roberts, 2009) based on real-world examples. Especially in light of the last two applications fields, inverse problems and reference data, sandbox provides the tool for creating virtual twins of sediment section, and hence, to define the problem solution as a basis for comparing the performance of competing or new analytical routines. This is mainly in times of evolving machine learning approaches essential as those powerful tools rely on well-defined and labelled training or reference data.

## 5.6 Limitations

The structure of the package was designed to allow for extensive flexibility and computational efficiency. This required setting a few fundamental assumptions, which resulted in structural limitations. Some of these limitations may be partly accounted for by workarounds. Most fundamentally, sandbox has no methods to account for erosional processes implicitly. As mentioned above (Sect. 5.1), the framework is based on a valid and intact age-depth relationship.

There is no support for post-depositional modification of grain properties at the moment. Such post-depositional dynamics may be added by defining further rules. For example, pedoturbation may be implemented as the probability to find grains from depths other than the actual sampling depths for each grain in a sample container, i.e., a rule that says if one sample at 5 m depth there is a 10 % chance to sample grains from 10 cm below.

The 1D structure is another structural limitation of sandbox; wanted though. If in future the demand arises, the model can be expanded to 2D or 3D. However, this would come at the cost of defining rules not just for the depth direction but also in lateral directions. At present it is more feasible to tackle such scenarios by defining different virtual sediment sections.



There are no topologic relations among the sampled grains. Apart from depth information for each grain, `sandbox` can neither provide information on the 3D location of the grains within a sample container nor on the distances among their

centroids. This precludes asking questions that require grain-to-grain information.

## 6 Conclusions

The R package `sandbox` provides a flexible and scalable framework to tackle research questions emerging from palaeoenvironmental reconstruction and numerical landscape representation. Its structure and available functions allow creating a virtual twin of given or artificially designed sediment sections focusing on sediment grains and their properties along a depth vector.

The current focus on geochronology is a pragmatic one. The framework can be used for numerous further cross-discipline topics, including geochemical analysis, soil formation representation, inverse modelling, and reproducible reference data set generation.

## 7 Data and code availability

The `sandbox` source code will be made available as an R package on CRAN and source code and development transparently

accessible via GitHub (https://github.com/coffeemuggler/sandbox). The supplementary materials contain an extensive manual to the package and the code used to prepare the figures in the main text.

*Author contributions.* Michael Dietze designed the code and initiated the manuscript. Sebastian Kreutzer reviewed and optimised package code, linked RLumModel and contributed age data for the loess section rule book. Margret C. Fuchs advised on the early stages of the package and translated luminescence laboratory techniques to package functions. Sascha Meszner contributed all field-based sedimentological and

stratigraphic base data and provided the grain-size measurement data and interpretation. All authors shared responsibilities in writing the manuscript.

*Acknowledgements.* SK has received funding from the European Union's Horizon 2020 research and innovation programme under the Marie Skłodowska-Curie grant agreement No 844457 (CREDit).





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
