# Peer review of "sandbox – Creating and Analysing Synthetic Sediment Sections with R"

_Geochronology, 2021_

## Referee Comment (RC1)

**1 Strengths**

The lead author is one of the most accomplished `R` programmers in the Earth Sciences. The `sandbox` package is logically implemented and well documented. It follows `R`'s list-based programming paradigm and uses the language's strengths in data visualisation whilst avoiding some of its weaknesses. For example, the package uses parallel processing for its calculation, thereby overcoming `R`'s computational limitations. All this is achieved with minimal dependencies. Unlike many other packages, sandbox does not require `ggplot2`, which makes the package light and nimble.

**2 Major comments**

**2.1 Usefulness**

I am not sure how useful `sandbox` is, and I doubt that it will ever become very popular. The three applications provided in Section 3.5 are not very convincing. Take, for example, the first case study, which investigates the effect of sample geometry on the single grain OSL age distribution. It shows that a cylindrical container results in a circular age distribution:

[Figure]

It is easy to derive this distribution analytically:

```r
x <- seq(from=-1,to=1,length.out=100)
y <- sqrt(1-x^2)
sedrate <- 1/8    # cm/yr, sedimentation rate
thickness <- 1    # cm, diameter of the container
t0 <- 25711       # age in centre of the cylinder
age <- x*thickness/sedrate + t0
freq <- y/sum(y) # normalise
plot(age,freq,type='l')
```

[Figure]

Figure 1: Expected age distribution for a cylindrical container.

The second application is similar to the first, whereas the third example did not make sense to me. I did not understand why sieving would cause 'age inversions' in the depth profiles. Do the coarse grains contain more inheritence than the small ones? If this is so, then I must have missed where this was specified. This should be explained better.

**2.2 Crudeness**

The virtual sediment sections defined by `sandbox` 'rule books' are not process-based. The algorithms are purely statistical and do not include any physics. This severely reduces their degree of realism, and limits their usefulness, as I will explain next.

Each location in the virtual sediment section is assumed to contain a discrete number of subpopulations. Each of these distributions is assumed to follow a lognormal grain size distribution with corresponding normally distributed mineral densities and grain packing densities. To generate virtual samples, random numbers are selected from these distributions. The grain size, mineral density and packing density are chosen independently, assuming zero covariance between their respective (log)normal populations. I think that this is an unjustifiable oversimplification.

In real sedimentary sections, grain size, density and packing density are strongly correlated with each other. For example, Stokes' Law dictates that small zircon grains ($\rho = 4.65$ g/cm$^3$) are 'hydraulically equivalent' with larger quartz grains ($\rho = 2.65$ g/cm$^3$). Therefore, well sorted sediment exhibits a *size shift* between quartz and zircon. In a sand that contains both quartz and zircon, the zircon will tend to fill the gaps between the sand grains, thereby increasing the packing density. Although I am not an expert in OSL, I do think that this is important because zircon tends to be rich in actinides, and so the relationship between zircon and quartz affects the dose rate. Wouldn't this be a more important problem to simulate than the geometry of the sample container?

In summary, hydraulic sorting and selective entrainment impose a strong covariance structure on the physical properties of sediments, which `sandbox` currently does not capture. In principle, it is possible to embed this covariance structure into the `sandbox` package using multivariate (log)normal distributions. However, in practice, this would not be so easy to implement, because it would dramatically increase the number of parameters that need to be set in the rule books. A process-based algorithm would fix this, but it would require a complete redesign of the package. Unfortunately, I can't think of a third solution.

**3   Other comments**

1. It is not clear from the title why this paper was submitted to Geochronology. It is only in Section 3.5 (line 335) that the geochronological relevance of the sandbox package becomes apparent. This section uses the method to create some virtual OSL samples. This should be changed, especially because I am doubtful that sandbox will ever be used for any other applications. Suggestion: change the title of the paper to:

   "*sandbox – Creating and Analysing Synthetic OSL samples with R.*"

2. All the random samples in the `sandbox` package are drawn from (log)normal distributions, which are extracted from real datasets using Dietze and Dietze's `EMMAgeo` package. An alternative and more flexible approach would be to draw random numbers from any cumulative distribution. For example:

```
set.seed(1)
phi <- c(1,3,3.5,4,5,5.25,6,8)
cdf <- c(0,0.1,0.3,0.5,0.6,0.8,0.9,1)
fun <- splinefun(x=cdf,y=phi,method="monoH.FC")
nr <- 20
r <- runif(nr)
rgs <- fun(r) # random grain sizes
par(mfrow=c(1,2))
plot(x=phi,y=cdf,type='p')
y <- seq(from=0,to=1,length.out=50)
lines(x=fun(y),y=y)
matlines(x=rbind(rep(0,nr),rgs),y=rbind(r,r),lty=3,col='black')
matlines(x=rbind(rgs,rgs),y=rbind(rep(0,nr),r),lty=3,col='black')
plot(density(rgs),main='')
rug(rgs)
```

[Figure]

Figure 2: Left: drawing 20 random samples from an arbitrary cumulative grain size distribution (circles mark the anchor points). Right: kernel density estimate and rug plot of the 20 random grain sizes.

3. The paper is too long. `sandbox` is based on an inherently simple idea that I am confident could be explained in a paper half the length of the current manuscript. I enjoyed reading the example code and the supplementary information item. However, I must confess that I found the main text a bit tedious to get through.

---

## Author Response (AR1)

We thank both referees for their assessment of the manuscript and the constructive and thought provoking suggestions. We have implemented changes were needed and useful and explain in this letter where we have implemented changes and why. Please find below the comments and the corresponding replies.

Referee 1

1 Strengths

The lead author is one of the most accomplished R programmers in the Earth Sciences. The sandbox package is logically implemented and well documented. It follows R's list-based programming paradigm and uses the language's strengths in data visualisation whilst avoiding some of its weaknesses. For example, the package uses parallel processing for its calculation, thereby overcoming R's computational limitations. All this is achieved with minimal dependencies. Unlike many other packages, sandbox does not require ggplot2, which makes the package light and nimble.

Reply: We are thankful for the praise. Indeed, 'sandbox' is developed to run with minimal burden to be slim yet using mainly native R code. Yes, we decisively avoided chart junk overhead.

2 Major comments

2.1 Usefulness

I am not sure how useful sandbox is, and I doubt that it will ever become very popular. The three applications provided in Section 3.5 are not very convincing. Take, for example, the first case study, which investigates the effect of sample geometry on the single grain OSL age distribution. It shows that a cylindrical container results in a circular age distribution:

Reply: We made the examples decisively crisp and easy to follow in this introductory article. It is correct that sample container geometry effects (also in combination with different deposition rates) could be studied with less overhead code than required with 'sandbox'. The main purpose of this example is to document the general way of using 'sandbox'. There is always the chance to develop more elaborated, sophisticated and insightful examples. But when the first order goal is to let a reader follow the processing steps, then the deeper usefulness of the result may not necessarily dictate the example's scope.

That said, we understand the general criticism and have now explicitly explained in the text why we use simple examples instead of elaborated ones pursuing a non-trivial research question (last paragraph of the introduction, first paragraph of section "Application examples and parameterisation"), and discuss that example 1 could also be solved more directly (first paragraph of discussion section "Geometric sampling effects"). Hence, we cannot estimate today, whether or not 'sandbox' will become very popular on the long run, however, for us the article is a start on which we can further build on.

The second application is similar to the first, whereas the third example did not make sense to me. I did not understand why sieving would cause 'age inversions' in the depth profiles. Do the coarse grains contain more inheritance than the small ones? If this is so, then I must have missed where this was specified. This should be explained better.

Reply: We have clarified the manuscript to better point at the underlying concept that we do indeed allow grains of different populations (hence size) to have a different inherited age and that their mixture with depth dependent population contributions can cause age inversions, even amplified through sieving if that sieving selectively enriches those populations with high inherited ages (section "Application examples and parameterisation").

2.2 Crudeness

The virtual sediment sections defined by sandbox 'rule books' are not process-based. The algorithms are purely statistical and do not include any physics. This severely reduces their degree of realism, and limits their usefulness, as I will explain next.

Reply: This is statement correct, but intended by us. The package 'sandbox' is supposed to provide a general framework not a process-based sedimentation model. However, it allows to impose any physically based relationship and thus turn the model exercise into one that is constrained by physical "rules" of how the virtual section should be built and behave. We have clarified the manuscript and mention this concept now explicitly (first paragraph of section "Philosophy and structure of sandbox").

Each location in the virtual sediment section is assumed to contain a discrete number of subpopulations. Each of these distributions is assumed to follow a lognormal grain size distribution with corresponding normally distributed mineral densities and grain packing densities. To generate virtual samples, random numbers are selected from these distributions. The grain size, mineral density and packing density are chosen independently, assuming zero covariance between their respective (log)normal populations. I think that this is an unjustifiable oversimplification. In real sedimentary sections, grain size, density and packing density are strongly correlated with each other. For example, Stokes' Law dictates that small zircon grains ($\rho$ = 4.65 g/cm3) are 'hydraulically equivalent' with larger quartz grains ($\rho$ = 2.65 g/cm3). Therefore, well sorted sediment exhibits a size shift between quartz and zircon. In a sand that contains both quartz and zircon, the zircon will tend to fill the gaps between the sand grains, thereby increasing the packing density.

Reply: This is an excellent example to illustrate that 'sandbox' is not an all-case physically meaningful representation of sedimentation processes by default. It is correct that many grain (population) parameters are linked by chemical or physical properties and processes. However, as mentioned above, we decisively designed 'sandbox' to be like that. Meaning that by default 'sandbox' does not relate, for example grain size and specific density to packing density, does not mean that one cannot implement such a relationship.

We acknowledge this argument and have added several examples to the SI. There, we show how one could include specific density driven grain-size differences and the resulting packing density effects to a parameterisation of 'sandbox'. However, we think that including this rather elaborated example to the main manuscript would counteract our arguments made above (reply to point 2.1).

Although I am not an expert in OSL, I do think that this is important because zircon tends to be rich in actinides, and so the relationship between zircon and quartz affects the dose rate. Wouldn't this be a more important problem to simulate than the geometry of the sample container?

Reply: It is correct, the high concentration, in particular of U, in zircon grains might affect the dose rate to a level of which it may take over the role of a significant contributor causing strong dose-rate heterogeneities. However, compared, for instance to feldspar, representing a strong beta-emitter due to its potassium concentration, the effect is not of greater relevance in most sites where the amount of zircons is less than 1% of the overall sediment budget and usually well mixed within the sediment matrix. Moreover, if zircons are important they tend to accumulate in particular layers. In other words, while zircon related dose-rate effects on OSL ages may become more relevant than sample container shape effects in particular cases, the introduction of such a parameterisation example would inflate the manuscript. To compromise, we mention the zircon dose-rate effect as a motivation for a further potential application field of 'sandbox' (second paragraph of discussion section "Sample population effects").

In summary, hydraulic sorting and selective entrainment impose a strong covariance structure on the physical properties of sediments, which sandbox currently does not capture. In principle, it is possible to embed this covariance structure into the sandbox package using multivariate (log)normal distributions. However, in practice, this would not be so easy to implement, because it would dramatically increase the number of parameters that need to be set in the rule books. A process-based algorithm would fix this, but it would require a complete redesign of the package. Unfortunately, I can't think of a third solution.

Reply: We may argue for another solution. In the light of the correctly identified additional need to a) implement and b) parameterise one out of a considerable range of physical laws of sedimentation, which may be adequate in some but certainly not in all use cases, why not exemplarily show the basic steps of how to do this if needed and otherwise keep the flexibility of 'sandbox' to let users chose if and/or which physical relationships they want to use for their rule book? By following that latter suggestion, the effort of excessive parameterisation is separated from the 'sandbox' model definition and moved to external duties, as illustrated in the supplementary information.

In summary, we did the following to solve the raised issue: i) explicitly mentioning in the manuscript that realistic representations of sedimentary deposits may require the implementation of further physical relationships (second paragraph of section "Limitations"), and ii) work through its implementation in the supplementary information.

3 Other comments

3.1. It is not clear from the title why this paper was submitted to Geochronology. It is only in Section 3.5 (line 335) that the geochronological relevance of the sandbox package becomes apparent. This section uses the method to create some virtual OSL samples. This should be changed, especially because I am doubtful that sandbox will ever be used for any other applications. Suggestion: change the title of the paper to: "sandbox – Creating and Analysing Synthetic OSL samples with R."

Reply: We decisively picked Geochronology because the overarching theme of 'sandbox' is age-depth-parameter relationships of sedimentary deposits. This is a topic of paramount relevance for dating applications but often not exploited with the needed care. This becomes even more true when effects of the grain size structure are considered as systematic uncertainties that go into, for instance, Bayesian modelling frameworks, such as 'BayLum'.

The only true alternative to GChron (excluding non-open access journals) would be GMD. However, we are afraid that the circle of people we want to reach and inspire by 'sandbox' will not enthusiastically follow GMD as outlet of papers. When adding parameters like major or minor elements, cosmogenic nuclide concentrations as well as carbon fractions or other biomarker content of grain populations, we believe that 'sandbox' may well become an interesting tool for a wide range of scientists occupied with other than luminescence-dating routines.

3.2. All the random samples in the sandbox package are drawn from (log)normal distributions, which are extracted from real datasets using Dietze and Dietze's EMMAgeo package. An alternative and more flexible approach would be to draw random numbers from any cumulative distribution.

Reply: Parameters (and rules) can be defined using not just normal distribution functions, but also uniform, gamma and exact, as described in the manuscript and package documentation. We initially had the option to use empirical distribution functions, as well. However, the overhead and parameterize hassle was significant, so we decided (for now) to skip this additional option, assuming that gamma, normal, uniform and exact (no scatter) already provide quite a bit of freedom to define parameters. We should point out though that the grain-size definition indeed is in phi scale. This does limit flexibility, but only for this single parameter. However, that constraint was decisively made, in agreement with long standing support from sedimentology.

To clarify, EMMAgeo does not necessarily produce log-normal end-members but would in principal return any distributions underlying a mixed data set. In a recent paper (Dietze et al., 2021 DOI: 10.1111/sed.12929) this has been tested with synthetic data.

3. The paper is too long. sandbox is based on an inherently simple idea that I am confident could be explained in a paper half the length of the current manuscript. I enjoyed reading the example code

and the supplementary information item. However, I must confess that I found the main text a bit tedious to get through.

Reply: We shortened the original manuscript by 16 % (but had to add some requested text, yielding a total reduction by 12 %), a point also raised by referee two.

Referee 2

Summary: I think this is an interesting paper that makes a good start at addressing something that, as the authors point out, is not well addressed anywhere else. I am supportive of publication in something resembling its present form.

1. Broader comments:

-- Note: I should not be trusted as having adequately reviewed the R code. I rarely use R.

-- The authors motivate this work by pointing out that landscape evolution and sediment transport modeling projects tend to completely avoid most aspects of sediment accumulation. Of course avoiding this is sensible in lots of applications, because if you have a 2-d landscape evolution model, once you start accounting for a sediment pile that is not perfectly mixed, then you have to add an additional dimension, but only in some of the model cells, etc. But it is true that this is a limitation in a lot of applications, for example in simulating cosmogenic-nuclide concentrations in fluvial sediments leaving a basin that contains both eroding and accreting areas. So I am very supportive of putting some effort into this area.

Reply: We thank the referee for the supporting statement, also advertising that there is never or always a good time to start using R.

-- I agree with the other reviewer that this is a lot of software, and a lot of paper text, for a fairly underwhelming set of example applications. However, I can think of lots of other theoretical potential applications. In my own field, this could interestingly be applied to the issue of grain-size dependence of cosmic-ray-produced or fallout radionuclides in sediment, which produces results that are biased by sample preparation or grain-size selection. If you imagine trying to simulate this in a landscape evolution model for a watershed, one would have time steps in which packages of sediment with certain properties were deposited, and then later time steps in which packages of sediment were eroded in different increments, such that an eroded increment was a mixture of different previously deposited increments. The software described here is not directly designed for this application, obviously, but it thinks about the problem in a relevant way. Thus, I don't think the fact that the worked examples are fairly simple is necessarily a problem for the paper. I think it is OK, in fact, probably good, to present what one thinks is a fairly versatile tool even if one is not completely sure what to do with it.

Reply: That latter statement was our motivation to develop 'sandbox': to not be restricted to one concrete model tailored to a specific task but to have a scalable and modifiable one, to be able to approach different discipline's questions with a common technique. That said, we see the need – also in agreement with referee one – to remove the bias of too

few examples and too much overhead text. Hence, we have added a series of less trivial examples to the SI to allow the reader to better judge the scope of 'sandbox' and the effort and workflow to pursue further exemplary research questions. Also in line with referee one, we have shortened the text by 12 %.

-- Although it is true that this is basically a mixing model for sediment properties that don't necessarily have anything to do with geochronology *per se*, it is clear from the OSL example, and the fact that I immediately thought of another example having to do with cosmogenic-nuclide applications, that this is likely to be at least somewhat interesting for geochronologists. I can think of general examples in other areas of geochronology as well, for example having to do with paleomagnetism of sediments or bulk radiocarbon dating. So this may be a little off topic for Geochronology, but I think it is OK.

Reply: We welcome the additional possible examples from other than OSL related geochronology research that might fall into the application scope of 'sandbox'. We have briefly added some of them to the short list already mentioned in the discussion.

-- One general issue in reviewing papers about software is that really the point of the review is just to ensure that the paper is a good description of the software. Of course, all reviewers have a strong desire to suggest additional features of the software, or to complain that their favorite features are not included. However, it is not really a reasonable review criticism to demand that the software do things that the authors were not intending to do. From this perspective, I found the paper to be good. Even though I don't think in R, the paper fairly clearly describes how the software works and it is possible to get a good idea of its capabilities. I do agree with the other reviewer that some aspects of the introductory text (before the code examples) are longer than necessary and hard to understand. In this section, the authors describe the software design in very general and symbolic terms ("sandbox has a parametric and probabilistic design"...the paragraph near line 80 is particularly difficult) that are very hard to understand before the examples. Once the authors get to the examples, many of these points become immediately clear, which makes the introductory text appear more confusing. I strongly suggest that the authors remove some of this general introductory material and, perhaps, replace it with a simpler and clearer statement of the input and output parameters of the software ("Input parameters to the software are a set of properties, including the age, contributions of certain grain size components, etc. that are assumed to vary with depth in the section. The outputs are...."). Alternatively, the authors could start with the examples and then later explain how the example applications are special cases of more general design properties. At present, I am worried that many readers may not make it through the more abstract descriptions at the beginning. I became much more interested in the paper when I got to section 3, but if I were not reviewing it I might not have gotten there.

Reply: We have revised the text and shortened the introduction, clarified and shortened the parameter description, and raise the importance of the examples early on. See also replies to requests by referee one. However, we believe that the section "Philosophy and structure of sandbox" is an important one because the model differs substantially from

other numerical approaches and therefore a general provision of background is relevant for readers.

There are, however, two missing aspects of the software that I think are important to discuss in the context of possible geological applications. The first has to do with age models. The other reviewer alluded to this in pointing out that failing to allow for correlation between various properties of a sediment section is oversimplified in relation to real life. I agree with this, but am more concerned about oversimplification of the age model. The assumption that the age model is linear between specified (age,depth) points is quite important in many applications in which one might want to know the properties of a large sample that spans a range of ages. In reality, sediment accumulation is not expected to be linear, but rather episodic and commonly autoregressive. Thus, expected internal variability in accumulation rates would significantly change, e.g., the distribution of OSL grain ages in a thick sample. As the software is designed, it appears that it would only be possible to simulate this effect by generating a large number of age-depth models with additional synthetic age/depth pairs between the known ones, that had been generated to have whatever autoregressive (or not) properties that you want, and repeatedly generating and sampling synthetic sections for each one of the age models. I suppose, therefore, that the software doesn't really oversimplify this aspect -- it just leaves it as an exercise for the student -- but I think it would be helpful to have some discussion of this issue in the paper.

Reply: We have revised the text regarding that latter aspect, simplistic representation of autoregressive accumulation processes (for the former, correlation of parameters, see next reply). Thereby, we also had to take care to not further inflate the text and add more complicated examples with more introductory overhead. Thus we decided to explain that 'sandbox' can be run in a very simple way, using linear accumulation rules, but that it can also be used with more realistic, complex parameterisation of the age-depth relationships (supported by a more elaborated example in the SI).

The second issue also to some extent echoes the other reviewer's point that there is no consideration of correlation between parameters, but the way I would express this is instead to point out that these correlations are often the result of physical relationships that must be honored for the synthetic sediment section to be a correct representation of the real one. The main example that stood out to me is that it appears that grain size distributions and density are set independently. In real life, on the other hand, for dry sandy sediments that typically achieve their closest possible packing at or immediately after deposition, the density is fully physically determined by the grain size distribution, so density is a completely dependent variable. A more poorly sorted sediment with a larger range of grain sizes always has a higher density (and also higher bulk strength properties...this is why a material with a larger range of grain sizes is called 'poorly sorted' in geology but 'well sorted' in civil engineering). I am not really an expert on this, but I do know that there is quite a lot of engineering literature relating grain size distribution to density, so it would be possible to incorporate a deterministic density calculation based on specified grain sizes. Again, this is getting pretty close to what I shouldn't be doing, which is to demand more features, but this aspect appears to me to be quite important as a potential reason that the synthetic sediment section might be a bad model for a real one.

Reply: In a similar way to the above point, we do now explain that 'sandbox' does not automatically come along with a dedicated set of physical rules that couple grain properties in an accumulated sediment sequence. We also mention that this apparent lack of coupling between parameters is due to the imposed flexibility, rather than an unwanted limitation. For application scenarios where these physical relationships among grain properties are not required, they would just add the burden of long parameterisation workflows. However, where such relationships are needed, they can be implemented in a rather flexible manner. We show this possibility for one dedicated particle accumulation law in the SI.

As a side note, the packing density only becomes relevant when a sample is generated because that parameter controls the number of grains than can be contained in a sample of a given volume. However, as explained in section "Philosophy and structure of sandbox", the 1D concept does not allow to depict grain-to-grain relationships and thus grain-size dependent void filling effects. Nevertheless, this is an important point and we do now mention this in the article (Fourth paragraph of section "Limitations").

2. Minor comments:

-- The authors should critically read the paper with an eye toward removing jargon that is not in standard usage and/or more confusing than necessary. For example, "geo-archive" provides no more information, and more confusion, than "sediment section." Suggest removing it. Another example is 'paleo-research.' Also confusing: if the paleoenvironment is the environment that existed in the distant past, is 'paleo-research' likewise research that was carried out in the distant past?

Reply: We are thankful for these hints and have removed the mentioned and further examples of unnecessary jargon.

-- I suggest that the authors remove a couple of sections in which they become somewhat opinionated about the desired properties of geoscience software. For example, the remarks in line 45-50 are likely true, but are not at all relevant to the point of the paper. Perhaps the way to think about this is that facts about the software being described ("This software is written in a language that runs on all platforms") are certainly important and appropriate, but general opinions about what the authors think software should be like ("Ideally, such a model is transparent and flexible throughout...") are off topic and better suited to a proposal or opinion article. There is another example in line 59, which sounds like a TV advertisement for R.

Reply: Well spotted! We anyway never got paid for it, so we now tuned into the science channel.

-- I didn't understand the sentence in 67-68. This seems to be an example of the general point above in which a very abstract description is very hard to follow before any examples are given.

Reply: The sentence has been removed during revision of the paragraph

-- While looking at a black-and-white printed copy of Fig. 4, it occurred to me that this could just be presented as a contour map, which would be simpler, not require color, and not require a legend.

Reply: Contours were added.